# Terahertz Displacement and Thickness Sensor with Micrometer Resolution and Centimeter Dynamic Range

**DOI:** 10.3390/s19235249

**Published:** 2019-11-29

**Authors:** Dae-Hyun Han, Lae-Hyong Kang

**Affiliations:** 1Department of Mechatronics Engineering, and LANL-JBNU Engineering Institute-Korea, Jeonbuk National University, 567 Baekje-daero, Duckjin-gu, Jeonju-si, Jeonbuk 54896, Korea; dh.han@jbnu.ac.kr; 2Department of Mechatronics Engineering, Department of Flexible and Printable Electronics, and LANL-JBNU Engineering Institute-Korea, Jeonbuk National University, 567 Baekje-daero, Duckjin-gu, Jeonju-si, Jeonbuk 54896, Korea

**Keywords:** centimeter dynamic range, displacement, sensor, terahertz, thickness, transparency

## Abstract

Measuring distance and thickness simultaneously is important in biological, medical, electronic, and various industries. Herein, we propose a method for simultaneously measuring the displacement and thickness of transparent materials using a pulsed terahertz wave. For this technique, a beam splitter was used to design the optical path such that the terahertz wave would incident the specimen vertically to achieve centimeter measurement range and micrometer resolution. The measured terahertz waveform carries peak time information reflected from the upper and lower surfaces of the sample, and the thickness can be calculated using the time difference between the first and second reflected peaks. The displacement can also be calculated using peak time difference when the sample moves from the initial position to the changed position. For validation, an experimental test was performed using aluminum, acrylic, and glass plates. The results confirmed a measurement range of 1 cm with an error of less than 23 μm, and the thickness error was less than 8 μm.

## 1. Introduction

Terahertz (THz) waves, whose portion of the electromagnetic spectrum extends from 100 GHz to 10 THz, have attracted extensive interest in recent years and have been widely used in a range of engineering fields [1,2,3,4,5,6,7], such as material characteristics analysis [1,2,4], testing of pharmaceutical tablet properties [2,5], and gas [6,7]. Compared with conventional sensing technologies, THz waves have numerous attractive spectral features associated with fundamental physical processes, such as a large amplitude in vibrational motions of organic compounds, rotational transition levels of molecules, lattice vibrations in solids, intra-band transitions in semiconductors, and energy gaps in superconductors [8]. One THz application is in simultaneously measuring the displacement and thickness of a sample (regardless of its transparency) in nonconductive materials. Non-contact displacement sensors [9,10,11,12,13] with micrometer resolution and centimeter operational range have been increasingly used for demanding measurement tasks involving highly sensitive surfaces to control precise movements or monitor target positions. A displacement sensor that uses a light source [9,10] is limited in measuring the displacement of transparent objects, but it can measure the spot region of a sample. Displacement sensors using eddy currents [11,12] can only measure displacement when measuring a sample on metal substrates, while ultrasonic displacement sensors [13] can measure displacement regardless of the substrate. Conversely, eddy current and ultrasonic displacement sensors have limitations in displacement measurement when they measure the spot region of a sample. Eddy current [14] and ultrasonic [15] thickness sensors used for engineering thick samples (above 100 μm) require direct contact between the sensor and sample surface. Most existing non-contact methods are capable of thickness measurements in the wavelength range of visible light and are effective with transparent solid-state and relatively stiff samples. However, it remains a challenge for full-field thickness measurements at the micrometer and centimeter dynamic range, regardless of the transparency of the materials. Another method for thickness measurements involves the use of THz waves. Terahertz-based thickness measurement methods have recently been introduced as a technology for measuring the thickness of various sample types. Yasui et al. [16] demonstrated a terahertz “paint meter” for non-contact thickness mapping of multilayer paints with relatively thick film builds (above 100 m). Yasuda et al. [17] proposed a numerical parameter fitting method that increases the sensitivity of the minimum thickness measurement. Su et al. [18] proposed the use of terahertz pulsed imaging as a novel tool for measuring the thickness and quality of car paint on both metallic and nonmetallic substrates. Han et al. [19] demonstrated a thickness measurement of the multi-delamination in glass-fiber-reinforced plastic (GFRP) using pulsed THz waves.

In this paper, a THz displacement and full-field thickness sensor is proposed to achieve high micrometer resolution and a centimeter dynamic range based on the excellent reflection and transmission properties of THz waves. The THz sensor was designed to propagate perpendicularly to the surface of the sample by employing a THz beam splitter. This configuration can achieve wider displacement and thickness measurement ranges of up to 10 mm and 5 mm, respectively. In the following section, we describe the principle and experimental setup of the method, followed by validation and an example application.

## 2. Measurement Principles and Setup 

The method reported herein for displacement and thickness measurement—regardless of the material transparency—is based on the characteristics of pulsed THz waves. In the case of the thickness measurement, only nonmetallic materials can be used. When pulsed THz waves are incident on a sample, reflections arise in the THz waveform due either to the interface between different mediums or to the extinction coefficient of metallic materials. The first reflected THz waves rise due to the interface between free space (air, $n_{air}=1$) and the sample surface. The refractive index of the sample is generally greater than that of air, so a positive peak is observed in the THz waveform. The negative peak is observed when the refractive index of the preceding medium is greater than that of the incident medium. In the case of metal surfaces, the positive peak is always observed due to the extinction coefficient of metallic materials [18].

### 2.1. Measurement of the Displacement and the Thickness

The time-resolved detection scheme of THz-TDS (Terahertz Time-Domain Spectroscopy) is directly applicable to measuring the displacement information of multilayer samples. When pulsed THz waves are incident on a nonmetallic sample, regardless of the transparency, the reflected THz waveform consists of a series of pulses reflected from the interface. The displacement of the sample can then be calculated in the THz reflection mode using the following equation:(1)$$D_{sample}=\frac{\left|t_{zero}-t_{sample}\right|}{2}\times \frac{c}{n_{air}}\times \cos\theta ,$$
where $D_{sample}$ represents the sample displacement, $t_{zero}$ is the peak time measured at the initial position of the sample, $t_{sample}$ is the peak time measured at the moving position of the sample, *c* is the speed of light in air, and $n_{air}$ is the refractive index of air. The thickness of the sample or defect can also be calculated in the THz reflection scan mode via the following equation:(2)$$T_{sample}=\frac{\Delta t}{2}\times \frac{c}{n_{sample}}\times \cos\theta ,$$
where $T_{sample}$ represents the sample thickness, Δt is the time between successive reflections, *c* is the speed of light in air, and $n_{sample}$ is the refractive index of the sample; the factor of one-half arises as the THz waves are measured in reflection mode.

### 2.2. Design of the THz Displacement and Thickness Sensor

The test setup and schematic diagram for the proposed THz displacement and thickness sensor is shown in Figure 1a,b, respectively. The sensor mainly consists of a femtosecond laser (T-Light, Menlo Systems Corp., Planegg, Germany) with a 1560 ± 20 nm wavelength and 100 ± 1 MHz repetition rate, optical delay unit (ODU; Menlo Systems Corp., Planegg, Germany), photoconductive antenna (PCA) for the emitter and detector, motorized linear stage (M-403.6PD, PI Corp., Auburn, MA, USA), data acquisition (DAQ; NI Corp., Austin, TX, USA), and THz optics including a TPX (Polymethyl Pentene) lens (for collimation or focusing) and beam splitter. The motorized linear stage is used to compare the performance of the THz displacement sensor and commercial displacement sensor based on a laser triangulation method in the centimeter range.

The optical path of the THz displacement and thickness sensor was designed such that the pulsed THz waves are incident perpendicular to the surface of the sample to achieve the centimeter measurement range, as shown in Figure 2. In brief, separated pulsed laser light with femtosecond pulse width supplies the THz emitter and a detector through the ODU. The emitted THz waves are collimated after passing through a TPX lens and then separated by a THz beam splitter, where 60% travels along the designed optical path and 40% enters the free space. The pulsed THz waves are focused on the sample surface after passing through a TPX lens. Subsequently, the THz waves reflected at the air–sample interface (such as the interface of different medium) are focused onto the THz detector. The reflected pulsed THz waves are collimated after passing through a TPX lens and reflected 90° toward the THz detector by the beam splitter.

## 3. Experiments and Applications

### 3.1. Displacement Measurement Range Performance Comparison 

The performance of the proposed THz displacement sensor was compared with an OD350 laser displacement sensor (range: 250 mm; resolution: 200 μm; SICK Corp., Waldkirch, Germany) for long-range measurement and with an LK-081 laser displacement sensor (range: ±15 mm; resolution: 3 μm; KEYENCE Corp., Osaka, Japan) for high accuracy measurement resolution with short measurement range. An aluminum plate was placed vertically on the motorized linear stage, and each THz displacement sensor was placed on the optical table, as shown in Figure 1a, to measure the displacement exerted on the sample. The displacement inputs were implemented within the range of 0–10 mm. The sample displacement was controlled using the motorized linear stage in 50 μm increments over a range of 10 mm. Next, the impact of sample transparency on sensor performance was evaluated by changing the sample material from an aluminum to a glass and acrylic plate.

Figure 3 shows the displacement measurement range results according to commercial Laser Displacement Sensor #1 (OD350, SICK Corp., Waldkirch, Germany), #2 (LK-081, KEYENCE Corp., Osaka, Japan), and the proposed THz displacement sensor. The proposed sensor produced measurement results similar to those of the commercial sensors. In the case of a displacement sensor using a laser source, it is impossible to measure the displacement of a transparent sample. However, the THz displacement sensor can measure not only the displacement of transparent materials (such as glass or acrylic plates) but also metallic materials (such as aluminum plates) with centimeter measurement range, as shown in Figure 4.

### 3.2. Measurement Accuracy Evaluation of the THz Displacement Sensor

The displacement of a manual-stage-mounted sample was controlled from 0 to 10 mm in 2.5 mm increments (0, 2.5, 5.0, 7.5, and 10.0 mm) to evaluate the measurement accuracy. At each position, the precision resolution of the THz displacement sensor was considered by increasing the displacement in 10 μm increments using a manual stage (5 μm resolution). The detailed conditions and comparison results are summarized in Table 1. Figure 5 shows the measured displacement and magnitude according to the displacement of the manual stage. The measured displacement linearly increased corresponding to the distance of the manual stage. The magnitude of the maximum peak was constant up to 2.5 mm and then decreased to 5 mm. The measurement error was calculated, and the results are illustrated in Figure 6. The distance error was less than ±10 μm based on the measured value—regardless of the sample material—for the entire measurement range, but the error of the measured displacement increased from 5 to 10 mm, as shown in Figure 6c–e.

The reason for this increase can be attributed to the depth of focus of the plano-convex lens. In this study, we used a plano-convex lens (TPX lens) with 50 mm focusing length. The depth of focus can be calculated using the following equation [20]:(3)$$2z_{0}=\;\frac{8}{\pi }\times \frac{c}{f}\times {\left(\frac{f_{L}}{D}\right)}^{2},$$
where $z_{0}$ represents the Rayleigh range (the depth of focus is two times the Rayleigh range), *c* is the speed of light in air (299, 792, 458 m/s), $f_{L}$ is the effective focal length (50 mm), and D is the diameter of the TPX lens. As such, the depth of focus can be calculated as 7.6 mm (@ 0.1 THz) to 0.76 mm (@ 1.0 THz). This means that the effective focusing distance is 46.2–53.8 mm @ 0.1 THz. As a result, not only did the magnitude of the peak point decrease, as shown in Figure 5, but also the error of the distance increased beyond the 5 mm distance, as shown in Figure 6.

### 3.3. Simultaneous Displacement and Thickness Measurement

One of the goals of this study was the simultaneous measurement of the displacement and thickness of a transparent sample. Three samples were used to verify the effect of displacement from the zero position on the thickness measurement: an aluminum plate, an acrylic plate of 5 mm thickness, and a glass plate of 1.42 mm thickness, as summarized in Table 2. The displacement of the samples was controlled from 0 to 2.5 mm. Figure 7 shows the measured THz waveforms according to sample material at 0 and 2.5 mm displacements of the sample. Figure 7a,c and e show the thickness of the sample at the 0 mm stage position, while Figure 7b,d,e show the same at the 2.5 mm stage position. When the displacement changed from 0 to 2.5 mm, the measured displacement was 2.503 mm and 2.496 mm for the acrylic and glass plate, respectively. The thickness of the acrylic plate changed slightly from 4.996 to 4.992 mm, while that of the glass plate changed from 1.420 to 1.417 mm. The thickness of the aluminum plate could not be measured due to the conductivity of the material. Based on a comparison of these results with those of a thickness gauge, the thickness measurement accuracy was found to be within 8 μm.

## 4. Conclusions

In summary, we have reported a new approach for measuring the displacement and thickness of transparent materials. We have shown that it is possible to measure the displacement (distance) of both metallic and nonmetallic transparent samples with centimeter range and micrometer resolution. The results of the terahertz displacement and thickness sensor were compared with those of commercial laser displacement sensors and a thickness gauge. The results showed a displacement measurement range of 1 cm with less than 23 μm error, and thickness measurement error of less than 8 μm. The errors of the measured displacement and thickness increased after 5 mm displacement due to the effective focal length of the TPX lens. This problem can be resolved by replacing the plano-convex lens with a longer focusing length.

The proposed terahertz displacement and thickness sensor has the following advantages: (1) non-contact measurement; (2) the ability to simultaneously measure the displacement and thickness of transparent samples, such as acrylic and glass plates; (3) the ability to measure the thickness of the individual layers of a multilayered sample; (4) the ability to provide thickness and displacement distribution maps for a sample, as well as single-point measurements. 

When THz imaging performed, maintaining the focal length is an important factor. However, if we use this result, it can be performed without an additional distance sensor to keep the focal length between the THz sensor and sample surface.

## Figures and Tables

**Figure 1 sensors-19-05249-f001:**
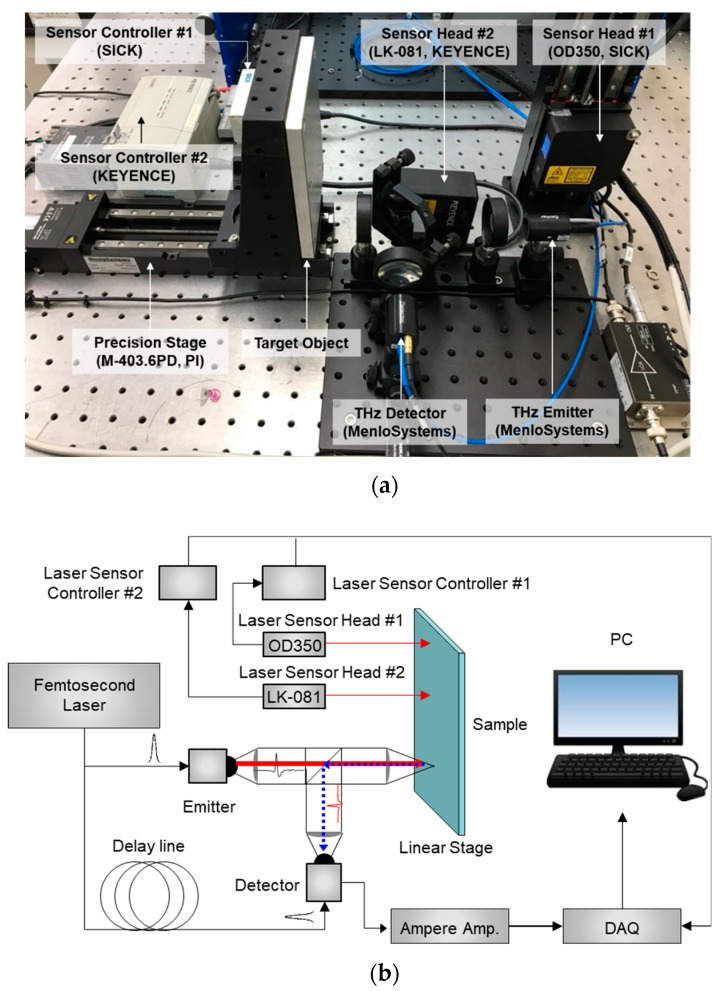
Terahertz (THz) displacement and thickness sensor: (**a**) experimental test setup; (**b**) schematic diagram.

**Figure 2 sensors-19-05249-f002:**
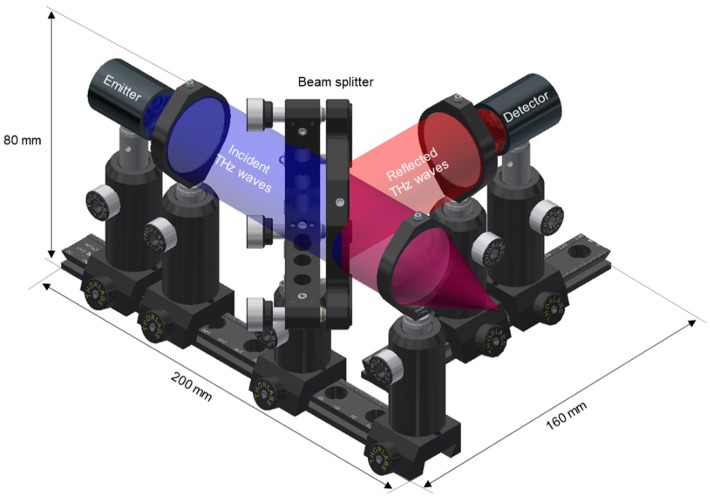
Designed optical path for the THz displacement and thickness sensor.

**Figure 3 sensors-19-05249-f003:**
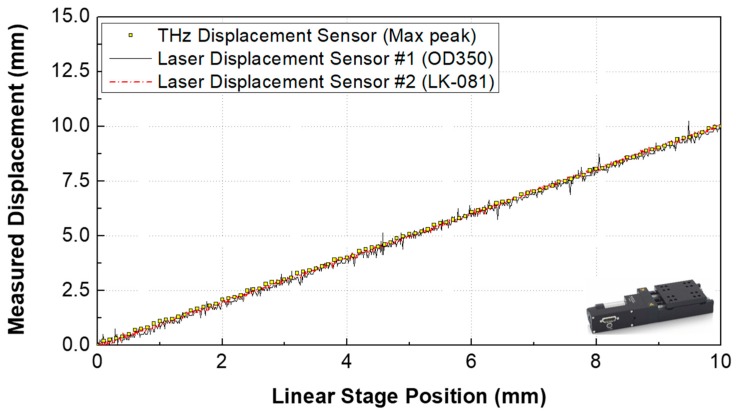
Displacement results according to the two laser displacement sensors and the proposed THz displacement sensor.

**Figure 4 sensors-19-05249-f004:**
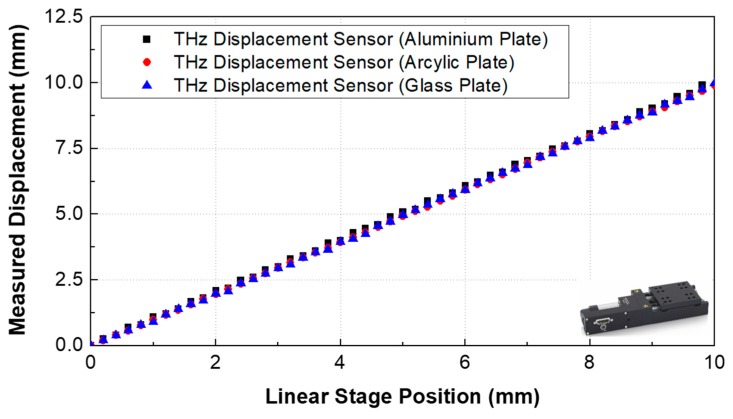
Displacement results for each target material using the THz displacement sensor.

**Figure 5 sensors-19-05249-f005:**
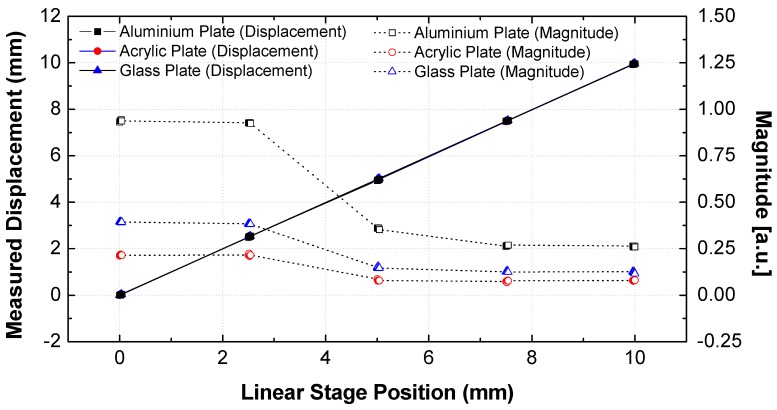
Measured displacement and magnitude of THz waveform according to target material.

**Figure 6 sensors-19-05249-f006:**
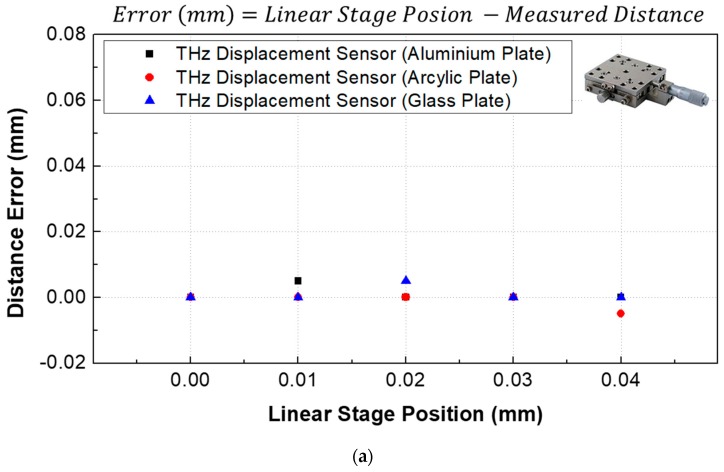
Measured displacement error according to target materials with centimeter measurement range: (**a**) 0.00–0.04 mm; (**b**) 2.50–2.54 mm; (**c**) 5.00–5.04 mm; (**d**) 7.50–7.54 mm; (**e**) 9.96–10.00 mm.

**Figure 7 sensors-19-05249-f007:**
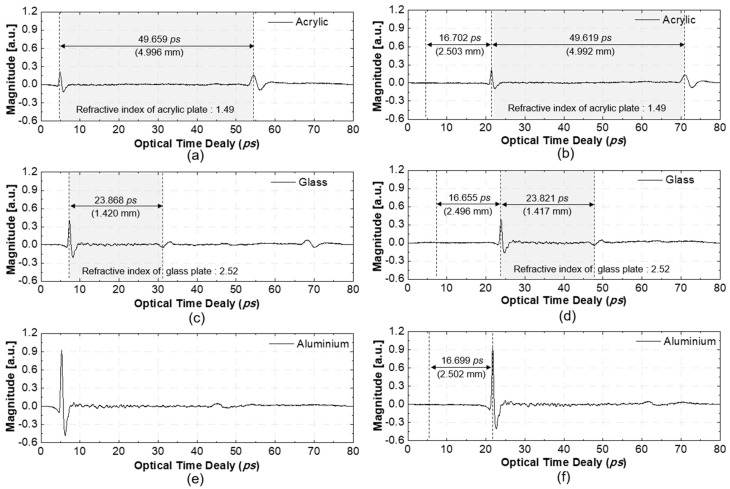
Measured Displacement and thickness according to the target materials: (**a**) thickness of the acrylic plate at zero position; (**b**) thickness and displacement of the acrylic plate at 2.5mm pistons; (**c**) thickness of the glass plate at zero position; (**d**) thickness and displacement of the glass plate at 2.5mm pistons; (**e**) a THz waveform of the aluminum plate at zero position; (**f**) displacement of the aluminum plate at the 2.5 mm position

**Table 1 sensors-19-05249-t001:** Measured displacement results using the THz sensor.

Stage Disp.(mm)	Measured Displacement (mm)			Displacement Error (mm)		
	Aluminum	Acrylic	Glass	Aluminum	Acrylic	Glass
0.00	0.000	0.000	0.000	0.000	0.000	0.000
0.01	0.005	0.010	0.010	0.005	0.000	0.000
0.02	0.020	0.020	0.020	0.000	0.000	0.000
0.03	0.030	0.030	0.030	0.000	0.000	0.000
0.04	0.040	0.045	0.040	0.000	−0.005	0.000
2.50	2.499	2.494	2.499	0.001	0.006	0.001
2.51	2.509	2.509	2.509	0.001	0.001	0.001
2.52	2.519	2.519	2.519	0.001	0.001	0.001
2.53	2.529	2.529	2.529	0.001	0.001	0.001
2.54	2.544	2.539	2.539	−0.004	0.001	0.001
5.00	4.939	4.959	4.984	0.017	0.017	0.017
5.01	4.954	4.969	4.994	0.017	0.017	0.017
5.02	4.964	4.979	5.009	0.017	0.017	0.012
5.03	4.974	4.989	5.014	0.017	0.017	0.017
5.04	4.989	4.994	5.023	0.017	0.012	0.017
7.50	7.483	7.478	7.483	0.022	0.022	0.017
7.51	7.493	7.488	7.493	0.022	0.022	0.017
7.52	7.503	7.498	7.503	0.017	0.022	0.017
7.53	7.513	7.508	7.513	0.022	0.022	0.017
7.54	7.523	7.518	7.523	0.022	0.022	0.017
9.96	9.927	9.932	9.937	0.033	0.028	0.023
9.97	9.937	9.942	9.947	0.033	0.028	0.023
9.98	9.947	9.952	9.962	0.033	0.028	0.018
9.99	9.957	9.967	9.967	0.033	0.023	0.023
10.00	9.962	9.977	9.977	0.038	0.023	0.023

**Table 2 sensors-19-05249-t002:** Measured thickness of the transparent materials using a THz sensor.

Target Materials		Thickness (mm)	
		THz Thickness Sensor	Thickness Gauge
Acrylic plate	at 0 mm	4.996	5.000
	at 2.5 mm	4.992	
Glass plate	at 0 mm	1.420	1.420
	at 2.5 mm	1.417	

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
