# Peer review of "Terahertz Displacement and Thickness Sensor with Micrometer Resolution and Centimeter Dynamic Range"

_sensors, 2019, doi:10.3390/s19235249_

Round 1

Reviewer 1 Report

In the manuscript “A terahertz displacement and thickness sensor with a micrometer resolution and centimeter dynamic range” authors present the ground work for distance and thickness measurements of thin plates with micrometer accuracy using THz waves generated with a femtosecond laser and a photoconductive antenna.

In general the paper is sound, with few issues on the scientific content. There is excellent agreement between the THz sensor and the commercial sensors.

I am suspicious that there is a small systematic error in the measurement or calculation since, when the stage displacement is greater than 2.54 mm, the errors are always positive, the measured thickness is always smaller than the real thickness.

There are many grammatical errors: missing articles, incorrect singular and plural, etc. A native English speaker should proof read the text before it is published.

Figure 2 needs a scale bar for the 3-D drawing. It would also be helpful to compare the size of the current setup to commercial setups not in the THz. It gives the authors motivation to continue their work and reduce the overall footprint.

Author Response

Thank you for your kind and valuable comments.

The answer to your comments is provided as an attached file.

Thanks.

Reviewer 2 Report

Comments for 643560

The authors present a displacement and thickness measurement method for transparent material using terahertz (THz) wave simultaneously. They are able to achieve a measurement range of 1 cm with less than 23 μm error (0.23%) and thickness error of less than 8 μm (0.08%).

I recommend the acceptance of this paper if the following minor questions are addressed.

Are there merits to split the THz wave to 60% and 40%? is this ratio optimized to achieve the best results? Are the errors presented contributed due to the limitation of the resolution of the manual stages? Will a more precise stage help to reduce the measurement error? Will the presented method continue to work when the material is absorptive? If so, is there a tolerance to how much the signal can be attenuated?

Author Response

(The authors gave the same response as above.)
